# Diagnosis and Management of Mitochondrial Encephalopathy, Lactic Acidosis, and Stroke-like Episodes Syndrome

**DOI:** 10.3390/biom14121524

**Published:** 2024-11-28

**Authors:** Ji-Hoon Na, Young-Mock Lee

**Affiliations:** Departments of Pediatrics, Gangnam Severance Hospital, Yonsei University College of Medicine, Seoul 06229, Republic of Korea; jhnamd83@yuhs.ac

**Keywords:** mitochondrial encephalopathy lactic acidosis and stroke-like episodes, MELAS, mitochondrial disease, *MT-TL1*, stroke-like episode

## Abstract

Mitochondrial encephalopathy, lactic acidosis, and stroke-like episodes (MELAS) syndrome is a complex mitochondrial disorder characterized by a wide range of systemic manifestations. Key clinical features include recurrent stroke-like episodes, seizures, lactic acidosis, muscle weakness, exercise intolerance, sensorineural hearing loss, diabetes, and progressive neurological decline. MELAS is most commonly associated with mutations in mitochondrial DNA, particularly the m.3243A>G mutation in the *MT-TL1* gene, which encodes tRNALeu (CUR). These mutations impair mitochondrial protein synthesis, leading to defective oxidative phosphorylation and energy failure at the cellular level. The clinical presentation and severity vary widely among patients, but the syndrome often results in significant morbidity and reduced life expectancy because of progressive neurological deterioration. Current management is largely focused on conservative care, including anti-seizure medications, arginine or citrulline supplementation, high-dose taurine, and dietary therapies. However, these therapies do not address the underlying genetic mutations, leaving many patients with substantial disease burden. Emerging experimental treatments, such as gene therapy and mitochondrial replacement techniques, aim to correct the underlying genetic defects and offer potential curative strategies. Further research is essential to understand the pathophysiology of MELAS, optimize current therapies, and develop novel treatments that may significantly improve patient outcomes and extend survival.

## 1. Introduction

Mitochondrial encephalopathy, lactic acidosis, and stroke-like episodes (MELAS) syndrome is a rare, maternally inherited disorder primarily caused by mutations in mitochondrial deoxyribonucleic acid (mtDNA), most notably the m.3243A>G mutation in the *MT-TL1* gene [1,2]. These mutations impair mitochondrial oxidative phosphorylation, leading to systemic manifestations, including stroke-like episodes, lactic acidosis, seizures, muscle weakness, and progressive neurodegeneration [3]. The pathophysiology of MELAS is complex, involving heteroplasmic mtDNA mutations that result in varying degrees of tissue dysfunction, particularly in energy-demanding organs such as the brain and skeletal muscles [2]. The clinical presentation can vary widely, with most patients experiencing early symptoms such as intractable seizures, diabetes mellitus, recurrent headaches, exercise intolerance, and sensorineural hearing loss [3,4]. Stroke-like episodes, a hallmark of the disease, often lead to rapid neurological deterioration and significantly contribute to morbidity and mortality [1,4]. Additional complications, such as cardiomyopathy and renal dysfunction, may also arise as the disease progresses [5]. Diagnosis is often confirmed through genetic testing, biochemical assays, and imaging studies. Despite advances in understanding its pathophysiology, MELAS remains a challenging disorder to manage, with current treatments focused primarily on symptom control. This multisystem disorder has a highly variable clinical course, reflecting the diverse nature of mitochondrial dysfunction and its impact on multiple organ systems [1,2,5,6].

MELAS is primarily caused by mutations in mtDNA, with the most common being m.3243A>G in the *MT-TL1* gene, which encodes transfer ribonucleic acid (tRNA) for leucine. This mutation disrupts mitochondrial protein synthesis, leading to impaired oxidative phosphorylation and subsequent energy failure in cells [7,8]. The m.3243A>G mutation is responsible for most MELAS cases, but other mtDNA mutations, such as those affecting complex I and tRNA genes, have also been identified, contributing to the clinical heterogeneity of the syndrome [9]. Heteroplasmy, a condition in which normal and mutated mtDNA coexist within cells, plays a critical role in the expression and severity of MELAS. The proportion of mutated mtDNA is directly correlated with the severity of clinical manifestations, as higher mutation loads typically result in more severe symptoms such as recurrent stroke-like episodes and progressive neurodegeneration [6,10]. This variability is also tissue-specific, with organs that have higher energy demands, such as the brain and muscles, being more severely affected when the mutation load surpasses a certain threshold. This phenomenon, known as the “threshold effect”, explains why MELAS can present with a wide range of symptoms across different patients, depending on the mutation load and the tissues affected [11]. Another key aspect of MELAS genetics is its maternal inheritance pattern. Mitochondria are passed exclusively through the maternal line; therefore, all offspring of an affected mother will inherit the mtDNA mutation, but only daughters will transmit it to the next generation [1,2,11]. Moreover, mtDNA mutations do not act alone; they interact with nuclear DNA, which encodes most mitochondrial proteins. This interplay between mtDNA and nDNA is crucial because disruptions in this interaction can exacerbate mitochondrial dysfunction and further impair energy production in cells [12]. In addition to the genetic complexity, MELAS also demonstrates the evolutionary and functional implications of mtDNA mutations because variations in mitochondrial function can affect energy production and cellular homeostasis [2,5,10]. These changes not only influence disease outcomes but may also impact long-term cellular adaptation, contributing to the broad spectrum of clinical phenotypes seen in patients with MELAS. The combination of mtDNA mutations, heteroplasmy, and nuclear-mitochondrial interactions complicates the understanding and management of MELAS [11,12].

Given the complexity of MELAS and its genetic underpinnings, accurate diagnosis and effective management remain significant challenges. This review focuses on the latest advancements in the diagnosis and management of MELAS, encompassing genetic testing, imaging techniques, and biochemical assessments that have enhanced scientists’ understanding of this disorder [13]. In addition, the review explores current therapeutic approaches, including symptomatic treatments and emerging experimental therapies targeting mitochondrial dysfunction [14]. By reviewing these advancements, this paper aims to provide a comprehensive overview of the current state of MELAS research, emphasizing the importance of further studies to address the remaining gaps in scientists’ understanding and improve patient outcomes.

## 2. Pathophysiology of MELAS

Mitochondrial dysfunction is the core driver of MELAS pathophysiology, predominantly caused by mtDNA mutations, with the m.3243A>G mutation in the *MT-TL1* gene being the most common. This mutation affects mitochondrial tRNA for leucine, thereby disrupting mitochondrial protein synthesis and impairing oxidative phosphorylation (OXPHOS). Consequently, adenosine triphosphate (ATP) production is reduced, which particularly impacts energy-demanding tissues such as the brain, skeletal muscles, and heart [15,16]. The m.3243A>G mutation compromises the function of the electron transport chain (ETC), which forces cells to rely on anaerobic glycolysis, thereby leading to the accumulation of lactate and lactic acidosis, a hallmark of MELAS [16,17]. In addition to *MT-TL1*, mutations in genes such as *MT-ND* that encode components of complex I in the ETC can also contribute to MELAS, illustrating the genetic heterogeneity of the syndrome [17,18]. A key factor influencing the severity of the disease is heteroplasmy, in which mutated and normal mtDNA coexist within cells. The proportion of mutated mtDNA, known as the “mutation load”, determines the severity of clinical symptoms, with higher mutation loads leading to more severe manifestations [17,19]. This variation in mutation load across tissues, especially in high-energy-demanding organs, accounts for the multisystemic presentation of MELAS [15,19]. Mitochondrial dysfunction also leads to an overproduction of reactive oxygen species (ROS), contributing to oxidative stress and cellular damage [18,19]. Additionally, impaired calcium homeostasis, particularly in neurons, results in excitotoxicity and apoptosis, thereby contributing to neurodegeneration [18,19]. The accumulation of mutated mtDNA over generations due to the lack of recombination in mitochondrial inheritance further exacerbates the progression of MELAS, which highlights the chronic nature of mitochondrial dysfunction [17,19]. These complex interactions between genetic mutations, energy failure, oxidative stress, and calcium dysregulation drive the progressive and multisystemic nature of MELAS. These nuclear gene mutations disrupt essential processes such as mitochondrial deoxyribonucleic acid (mtDNA) replication, transcription, and translation, which can exacerbate mitochondrial dysfunction and add another layer of clinical variability to the disease. The interplay between nuclear and mitochondrial mutations highlights the complexity of MELAS, making its diagnosis and treatment more challenging [20].

Stroke-like episodes (SLEs) in MELAS are complex, multifactorial events that significantly contribute to the morbidity and mortality associated with the syndrome. Unlike typical ischemic stroke episodes, these episodes are not caused by vascular occlusion but result from profound mitochondrial dysfunction that primarily impairs oxidative phosphorylation (OXPHOS) within neurons and astrocytes. The most common mutation associated with MELAS, the m.3243A>G mutation in the *MT-TL1* gene, leads to a significant reduction in ATP production and the subsequent accumulation of lactate. This metabolic imbalance creates a pathological environment where energy-demanding tissues, particularly the brain, are unable to meet their metabolic needs [21,22]. As a result, stroke-like lesions (SLLs) develop in areas of the brain that do not conform to typical vascular territories, distinguishing SLEs from ischemic strokes. These lesions are often seen in the parietal, occipital, and temporal lobes, and their progression is linked to episodes of neuronal stress caused by ATP depletion [1,23]. Mitochondrial angiopathy is another crucial component in the development of SLEs. The disruption of mitochondrial function extends beyond neurons and astrocytes, affecting the endothelial cells and smooth muscle cells of cerebral blood vessels. Mitochondrial abnormalities in these vascular cells impair the regulation of cerebral blood flow, often resulting in periods of hyperperfusion, followed by ischemic-like damage due to poor blood flow regulation [22,24]. Hyperperfusion during SLEs is paradoxical because it is driven by metabolic stress rather than by the occlusion of blood vessels, leading to vasogenic edema and contributing to the cortical damage observed during these episodes [21]. This dynamic alteration in cerebral blood flow plays a critical role in the recurrent nature of SLEs because repeated episodes lead to the gradual accumulation of irreversible damage, including cortical necrosis and gliosis.

Another pivotal mechanism in SLEs is the phenomenon of neuronal hyperexcitability, which is exacerbated by mitochondrial dysfunction. Neuronal and astrocytic failure to adequately clear excitatory neurotransmitters, particularly glutamate, from synapses leads to prolonged excitation, contributing to excitotoxicity. Astrocytes, which play a crucial role in maintaining the balance of excitatory neurotransmitters, become dysfunctional because of the lack of ATP, thereby impairing the uptake of glutamate from the extracellular space. This excitotoxic environment increases neuronal energy demand and places additional stress on the already compromised mitochondria [23,25]. Neuronal hyperexcitability is consequently a cause and a result of mitochondrial dysfunction, creating a vicious cycle that exacerbates the metabolic crisis during SLEs [6,22]. Recent evidence also highlights the role of nitric oxide (NO) in the pathogenesis of SLEs. Mitochondrial dysfunction in MELAS affects the production of NO, a critical regulator of vasodilation and blood flow in the brain. The depletion of NO disrupts cerebral perfusion, further complicating the balance between hyperperfusion and ischemia during SLEs [25]. NO deficiency, combined with preexisting mitochondrial and metabolic abnormalities, creates an environment in which the brain is highly susceptible to recurrent metabolic crises and structural damage [24,25]. The cumulative impact of these recurrent SLEs is profound. While some early changes observed in SLEs such as vasogenic edema may be reversible with prompt treatment, long-term effects such as cortical atrophy, laminar necrosis, and brain atrophy are permanent and progressive [16,21,25]. These structural changes are the result of repeated metabolic stress, excitotoxicity, and impaired blood flow regulation, all of which contribute to neurodegeneration in patients with MELAS [23,24,25].

The cellular mechanisms in MELAS syndrome involve intricate disruptions in mitochondrial function, with a particular focus on impaired oxidative phosphorylation (OXPHOS) and its downstream effects on cellular metabolism. A central aspect of this dysfunction is the decreased capacity of the mitochondria to produce adequate ATP, which is crucial for cellular energy, particularly in high-demand tissues such as the brain, muscles, and the cardiovascular system. This energy deficit triggers a cascade of detrimental processes, including the activation of anaerobic glycolysis, leading to lactic acidosis and further metabolic stress within affected cells [26,27]. In addition to energy production failures, mitochondrial dysfunction in MELAS contributes to an overproduction of ROS, which leads to oxidative damage to proteins, lipids, and DNA within cells. This oxidative stress exacerbates mitochondrial injury and contributes to the progression of neurodegeneration and multisystemic damage in patients with MELAS [9,27]. Mitochondria also play a key role in maintaining calcium homeostasis within cells. Dysfunction in this system leads to calcium overload in neurons, causing excitotoxicity and triggering apoptotic pathways, further accelerating cell death [9]. The inability of mitochondria to regulate these key processes results in progressive neurodegeneration, muscle weakness, and organ dysfunction. Additionally, defects in mitochondrial dynamics, including fission and fusion processes, contribute to the accumulation of dysfunctional mitochondria within cells. These processes are vital for maintaining mitochondrial health by segregating damaged mitochondria for repair or degradation. Impairments in these mechanisms lead to the accumulation of damaged mitochondria, amplifying the cellular stress response and worsening the energy crisis within cells [12,26]. Mitophagy, the process of removing defective mitochondria, is similarly impaired in MELAS, further contributing to the buildup of malfunctioning organelles [12,27]. This inability to effectively manage mitochondrial quality control exacerbates the chronic energy deficiency and oxidative damage that are hallmarks of the disease. Moreover, recent studies have highlighted the role of mitochondrial angiopathy, wherein small blood vessels, particularly in the brain, display abnormal mitochondrial proliferation within endothelial cells, leading to impaired blood flow regulation. This factor contributes to ischemic-like conditions in tissues, further compounding the energy deficit and contributing to the SLEs that are characteristic of MELAS [9,12]. These combined cellular and vascular mechanisms underpin the multisystemic nature of MELAS and highlight the complexity of its pathophysiology (Figure 1).

## 3. Clinical Presentation

Neurologic manifestations in MELAS are extensive and contribute significantly to patient morbidity and mortality. SLEs, a hallmark feature, primarily affect the occipital, parietal, and temporal lobes. These episodes present with transient or permanent deficits such as hemiparesis, aphasia, and cortical blindness, often evolving into progressive and irreversible neurological impairment [1,5,6]. The lesions caused by these episodes typically do not follow traditional vascular patterns and are more related to mitochondrial energy failure within neurons, which results in cytotoxic edema and neuronal necrosis. Over time, the accumulation of these episodes leads to cortical atrophy, exacerbating cognitive decline and motor deficits [28]. Magnetic resonance imaging (MRI) findings typically reveal transient diffusion abnormalities during the acute phase, which evolve into more permanent changes over time, including laminar necrosis and cortical atrophy. Neuroimaging often reveals widespread cortical and subcortical lesions in the brain, affecting particularly the occipital and parietal lobes. These lesions, combined with the involvement of the basal ganglia and thalamus, contribute to the complex and multifaceted neurologic presentation of MELAS [29,30]. These imaging patterns highlight the progressive and destructive nature of SLEs, contributing to long-term cognitive deficits and neurological decline [29,31]. Cognitive impairment is common and manifests as memory loss, decreased executive functioning, and diminished attention, stemming from mitochondrial dysfunction and from damage induced by recurrent SLEs [2,28,32]. Patients may also present with psychiatric symptoms such as depression and anxiety, which can be exacerbated by the progressive nature of the disease and the accompanying cognitive deficits [30,33,34].

Seizures are another critical neurologic feature in MELAS, affecting up to 90% of patients. Seizures can be focal and generalized or evolve into status epilepticus, significantly contributing to the overall neurological burden. The impaired energy metabolism in neurons caused by mitochondrial defects increases excitability, thereby leading to lower seizure thresholds and frequent severe epileptic events [32,35]. These seizures are often closely associated with SLEs and can further exacerbate neuronal damage, compounding the patient’s cognitive and functional decline over time [36]. Moreover, status epilepticus in MELAS can be resistant to treatment and is a major cause of morbidity [35].

In addition to SLEs and seizures, sensorineural hearing loss is a frequent and progressive manifestation in MELAS. Hearing loss tends to begin in adolescence and worsens progressively, often culminating in severe auditory deficits. This symptom may be caused by mitochondrial dysfunction within the cochlea and the auditory neural pathways, disrupting the energy-dependent processes of sound transmission and auditory perception [37]. Furthermore, patients with MELAS often experience peripheral neuropathy, which presents as chronic sensory and motor deficits in the extremities. Peripheral nerves, which are energy-intensive tissues, are especially vulnerable to mitochondrial dysfunction, leading to symptoms such as muscle weakness, numbness, and tingling [38]. This progressive neuropathy adds to a patient’s physical and functional disability, reflecting the widespread nature of mitochondrial dysfunction across multiple organ systems [2,5].

Cardiovascular complications in MELAS are a major source of morbidity and mortality, with hypertrophic cardiomyopathy being the most commonly observed cardiac manifestation. Hypertrophic changes in the myocardium are likely a result of energy deficits due to impaired oxidative phosphorylation, leading to myocardial hypertrophy as the heart attempts to compensate for the loss of efficient energy production [1,2,39]. Over time, hypertrophic cardiomyopathy can progress to heart failure, particularly when left ventricular outflow obstruction and diastolic dysfunction exist. Patients with MELAS are also at an increased risk for arrhythmias, including atrial fibrillation and ventricular tachycardia, because of the involvement of mitochondrial dysfunction in the conduction pathways of the heart [40]. Additionally, arterial hypertension is frequently observed, with mitochondrial dysfunction contributing to increased vascular resistance and endothelial dysfunction [41]. Studies suggest that mitochondrial abnormalities within the vascular smooth muscle cells and endothelial cells impair vasodilation, thereby promoting hypertension, which worsens cardiac workload and can lead to the earlier onset of heart failure [40,42]. Cardiovascular involvement in MELAS syndrome often manifests as hypertrophic or dilated cardiomyopathy, which may progress to heart failure. In one case, the successful simultaneous heart–kidney transplantation of a patient with advanced MELAS cardiomyopathy highlights the complexity of managing such severe cases. The patient, who had the m.3243A>G mutation, experienced heart failure because of biventricular cardiomyopathy. This finding emphasizes the risk of cardiovascular decompensation in patients with MELAS. Close cardiovascular monitoring and multidisciplinary intervention is a crucial need, especially because the progression to advanced heart failure can be rapid. Although transplantation is a potential option, it carries risks because of the multi-organ involvement of the disease [5,13,43].

Renal involvement in MELAS can manifest as a variety of nephropathies, including focal segmental glomerulosclerosis and tubulointerstitial nephritis, often leading to chronic kidney disease. The pathogenesis of renal dysfunction in MELAS is linked to mitochondrial dysfunction in the renal tubular cells, which are highly reliant on oxidative phosphorylation for energy. The inability of these cells to meet their energy demands leads to structural damage and progressive loss of renal function [1,44]. Additionally, mitochondrial defects in the glomeruli lead to proteinuria and nephrotic syndrome, further complicating the disease course [39,44]. The combination of renal and cardiovascular impairments poses a significant challenge to the management of these patients because the progression of one system’s dysfunction often accelerates the decline in another [43,44].

Endocrine abnormalities, particularly diabetes mellitus, are a significant feature of MELAS. MELAS is typically associated with insulin-dependent diabetes, which tends to manifest at a young age. The pathophysiology of diabetes in patients with MELAS is primarily due to mitochondrial dysfunction in pancreatic beta cells, which impairs insulin secretion [42]. This mitochondrial diabetes is closely linked to the m.3243A>G mutation, and patients often exhibit a unique clinical profile, including lower body mass index and earlier onset of the disease. The severity and onset of diabetes are correlated with the mutation load, with higher levels of mutated mitochondrial DNA leading to more severe insulin deficiency [45,46]. This form of diabetes can be challenging to manage because of its metabolic underpinnings, which differ from typical type 2 diabetes, and it often progresses more rapidly [45]. Additionally, the metabolic dysregulation exacerbates the overall energy deficit in MELAS, further increasing the burden on mitochondrial function and accelerating disease progression.

## 4. Diagnosis of MELAS

The diagnosis of MELAS requires a multidisciplinary approach that integrates clinical features, genetic testing, lactic acidosis in plasma and cerebrospinal fluid (CSF), and neuroimaging. However, the identification of pathogenic mitochondrial DNA mutations, particularly m.3243A>G, remains the cornerstone of the diagnosis. One of the most important diagnostic clues is the occurrence of SLEs in young individuals, typically under the age of 40, which are accompanied by other neurological symptoms such as seizures, headaches, and progressive cognitive decline. These episodes are not confined to a typical vascular territory, differentiating MELAS from other causes of stroke [1,2,47].

MELAS is diagnosed in a proband who meets the clinical criteria and has a pathogenic or likely pathogenic variant confirmed with genetic testing. Genetic testing plays a central and increasingly dominant role in confirming the diagnosis of MELAS. The m.3243A>G mutation in the *MT-TL1* gene, which encodes mitochondrial tRNA^Leu(UUR)^, is responsible for approximately 80% of all cases [47]. This mutation leads to impaired mitochondrial protein synthesis, which affects oxidative phosphorylation and leads to multisystem manifestations. Although m.3243A>G is the most common mutation, other pathogenic variants have also been identified, including mutations in the *MT-ND5*, *MT-TH*, and *MT-TK* genes [1,7,48]. Of note, the *MT-ND5* mutation is associated with a broader mitochondrial disease spectrum, but it also exhibits MELAS-like characteristics such as SLEs and progressive encephalopathy. *MT-TK* mutations, although rare, are linked with more severe phenotypes and rapid disease progression. Among these, m.13513G>A in the *MT-ND5* gene has been recognized in 10–15% of MELAS cases [1]. Other less common mutations, such as m.3271T>C and m.3252A>G in the *MT-TL1* gene, are also pathogenic variants contributing to the MELAS phenotype, albeit at lower frequencies [1,49].

The diagnostic process begins with targeted genetic testing for known pathogenic variants, typically starting with screening for the m.3243A>G mutation in the *MT-TL1* gene. However, in patients in whom this mutation is not detected, comprehensive analysis using next-generation sequencing or whole-exome sequencing is recommended to identify other less common mutations [50]. These techniques are highly sensitive and can detect low levels of heteroplasmy, in which normal and mutated mtDNA coexist within a cell. This finding is particularly important in MELAS because the heteroplasmic load can vary significantly between tissues, with higher levels existing in affected tissues such as muscle or in urine [1,50]. For example, m.3243A>G mutation heteroplasmy tends to be higher in muscle biopsies, compared with blood samples, which may explain the variability in clinical presentations among individuals [1,48]. If blood testing shows low heteroplasmy or is negative, testing of other tissues such as muscle biopsy or urinary epithelial cells may be required [1,50]. In prenatal settings, genetic testing of amniotic fluid or chorionic villi samples can provide early detection of MELAS mutations, although counseling regarding heteroplasmy and variable expression is essential [50].

In addition to identifying specific mutations, genetic counseling is crucial for patients and families with MELAS given the maternally inherited nature of mitochondrial DNA (mtDNA). Understanding the inheritance pattern and the implications of heteroplasmy can aid in predicting disease severity, as higher levels of mutated mtDNA are generally associated with more severe clinical manifestations. Furthermore, advances in genomic technologies are enabling more detailed analysis of mitochondrial mutations, allowing for better genotype–phenotype correlations, which may refine diagnostic criteria and improve disease management. For instance, m.3243A>G heteroplasmy levels of 50–70% are typically associated with diabetes and deafness, whereas higher levels (>80%) are linked with more severe neurological manifestations [1,19,48,50].

Muscle biopsy, although less frequently used because of advancements in genetic testing, remains a valuable tool, especially in patients in whom genetic mutations are not easily detectable. Histopathological examination of muscle tissue can reveal characteristic findings such as ragged red fibers (RRFs) on Gomori trichrome staining, which indicate abnormal accumulations of mitochondria within muscle cells. Additionally, electron microscopy can demonstrate abnormal mitochondrial morphology, further supporting the diagnosis [1,48].

MELAS clinically presents with a broad spectrum of features that extend beyond the neurological system. In addition to patients having the hallmark SLEs, they may experience recurrent headaches, seizures, muscle weakness, sensorineural hearing loss, and diabetes mellitus [5,50]. These symptoms are progressive, often resulting in severe disability and reduced life expectancy. Endocrine dysfunction, particularly diabetes, is commonly associated with the m.3243A>G mutation and can often serve as an early clinical clue toward the diagnosis [2,6].

Neuroimaging findings are indispensable in diagnosing MELAS because they reveal distinctive patterns not typically seen in other neurological disorders. MRI scans often show areas of hyperintensity on T2-weighted images, particularly in the parieto-occipital regions, but these areas do not correspond to typical vascular territories, distinguishing them from ischemic strokes [29,51]. SLLs often shift locations over time, a feature that is highly suggestive of MELAS. Along with MRI, magnetic resonance spectroscopy can detect elevated lactate levels in the brain, which are indicative of mitochondrial dysfunction and energy failure, further supporting the diagnosis [27,52]. The use of magnetic resonance spectroscopy (MRS) to detect lactic acid peaks in the brain is also a significant diagnostic tool. Elevated lactate levels are a marker of impaired oxidative phosphorylation and can be detected in both acute SLEs and chronic lesions. This metabolic signature is particularly useful when clinical and imaging findings alone are insufficient to confirm the diagnosis [52]. In addition, lactic acidosis in plasma and CSF serves as an indicator of systemic metabolic dysfunction and is an important finding for the clinical diagnosis of MELAS. Table 1 summarizes the diagnostic modalities and their clinical significance in MELAS (Table 1).

## 5. Management and Treatment Strategies of MELAS

The management of MELAS primarily revolves around symptomatic and supportive care, emphasizing a multidisciplinary approach to address the various organ systems involved. Given the multisystemic nature of the disease, coordinated care from neurologists, cardiologists, nephrologists, and endocrinologists is often required to manage the broad spectrum of clinical manifestations [5,6,32]. Vitamin and cofactor supplementation, particularly for mitochondrial support, plays a critical role in management. Coenzyme Q10, which acts as an electron carrier in the mitochondrial electron transport chain, supporting oxidative phosphorylation and stabilizing ATP production, typically dosed at 30 mg/kg/day, is commonly used to enhance mitochondrial function. Additionally, riboflavin (vitamin B2) at a dose of 50–400 mg daily may be beneficial, especially in patients with complex I deficiencies, to support complex I activity in the electron transport chain, thereby reducing oxidative stress and improving energy metabolism. L-carnitine, administered at 50–100 mg/kg/day, is frequently recommended to support energy metabolism. L-carnitine plays a vital role in the transport of long-chain fatty acids into the mitochondria for β-oxidation, a critical process for energy production. It enhances mitochondrial function by improving fatty acid metabolism and reducing the accumulation of toxic acyl compounds, which are common in mitochondrial dysfunction. While these treatments are not curative, they help manage symptoms and potentially slow disease progression [53,54,55].

More aggressive treatments include L-arginine and citrulline, taurine, and diet therapy [16,56,57]. These interventions aim to further support mitochondrial function and address specific metabolic deficiencies. L-arginine and citrulline supplementation can enhance NO production, potentially reducing the frequency and severity of SLEs [16]. Taurine, an amino acid, may aid in stabilizing cell membranes and improving mitochondrial efficiency [57]. Diet therapy, including the ketogenic diet (KD) or other specialized nutritional plans, can provide alternative energy substrates to bypass dysfunctional mitochondrial pathways, thereby supporting neuronal function and reducing metabolic stress [56]. Implementing these aggressive treatments requires careful monitoring and coordination among healthcare providers to tailor interventions to the individual patient’s needs and organ involvement (Figure 2).

### 5.1. L-Arginine and Citrulline Supplementation

L-arginine and L-citrulline supplementation is based on the essential role of NO in maintaining vascular homeostasis and cellular energy metabolism. NO, produced from L-arginine by endothelial nitric oxide synthase (eNOS), is a critical mediator of vasodilation and blood flow regulation. In MELAS syndrome, impaired mitochondrial function leads to endothelial dysfunction and decreased NO bioavailability, thereby contributing to the occurrence of SLEs. L-citrulline, a precursor to L-arginine, is particularly valuable as it enhances NO production by increasing L-arginine availability through the L-citrulline-L-arginine cycle, bypassing the first-pass metabolism in the liver [58,59,60,61]. This mechanism allows for more sustained NO synthesis and improved vascular function, particularly in the brain, where proper blood flow regulation is essential for preventing neurological damage during SLEs [59,62,63]. Clinical studies have demonstrated that L-arginine and L-citrulline supplementation can effectively reduce the frequency and severity of SLEs in patients with MELAS by improving endothelial function and restoring NO levels [59,62,64]. The increased NO production enhances cerebral blood flow, reducing oxidative stress and promoting better oxygen and nutrient delivery to energy-deprived tissues [62,65]. Furthermore, by improving NO bioavailability, these supplements help alleviate the metabolic crisis associated with mitochondrial dysfunction, thus reducing lactic acidosis and enhancing overall cellular energy metabolism [66]. In terms of dosage, L-arginine is typically administered during acute SLEs and as maintenance therapy in patients with MELAS. During an acute SLE, the recommended treatment is a bolus of intravenous L-arginine (500 mg/kg for children or 10 g/m² of body surface area for adults) within 3 h of symptom onset, followed by a continuous infusion of the same dose over 24 h for the next 3–5 days. This approach aims to restore NO levels and improve cerebral perfusion during the acute phase. For maintenance therapy, the prophylactic administration of oral L-arginine at a dose of 150–300 mg/kg/day, divided into three doses, is recommended to reduce the risk of recurrent SLEs [66,67]. This dosage provides sustained NO production and helps in long-term vascular protection and metabolic regulation.

However, as with all treatments, L-arginine and L-citrulline supplementation may have adverse effects. Gastrointestinal issues, including nausea, diarrhea, and bloating, are the most commonly reported adverse effects, particularly at higher doses of L-arginine. In rare cases, patients may experience low blood pressure (i.e., hypotension), which results from the vasodilatory effects of enhanced NO production. Additionally, long-term use of these supplements requires careful monitoring to avoid potential electrolyte imbalances such as hyperkalemia, which can occur because of altered renal function in some patients. Despite these risks, the overall benefit of supplementation in managing MELAS often outweighs the potential adverse effects when used appropriately and under medical supervision [66]. Thus, L-arginine and L-citrulline supplementation offer significant therapeutic potential in managing the complex vascular and metabolic abnormalities in patients with MELAS, particularly via their ability to enhance NO-mediated vasodilation and reduce the harmful effects of mitochondrial dysfunction on various organ systems [16,63].

### 5.2. High-Dose Taurine Therapy

High-dose taurine therapy has emerged as a promising treatment for MELAS, particularly for its potential to mitigate SLEs. Taurine, a sulfur-containing β-amino acid, plays a crucial role in mitochondrial function, notably by modifying the first anticodon nucleotide of mitochondrial tRNA^Leu(UUR)^, which is a critical component for decoding codons during protein synthesis. In patients with MELAS with the common 3243A>G mutation, this taurine modification is absent, thereby resulting in defective mitochondrial protein translation and impaired respiratory chain activity [68,69]. Clinical trials have demonstrated the efficacy of taurine supplementation in reducing the frequency and severity of SLEs in patients with MELAS. A multicenter phase III trial reported that high-dose taurine (9–12 g/day) significantly reduced the annual relapse rate of SLEs, with approximately 60% of participants achieving complete response [57]. Additionally, taurine has been shown to rescue mitochondria-related metabolic impairments, particularly in induced pluripotent stem cells derived from patients with MELAS. This rescue effect is largely attributed to taurine’s ability to reduce oxidative stress and restore glutathione levels, which are critical for counteracting mitochondrial dysfunction in MELAS [70]. Taurine mechanistically plays an important role in enhancing mitochondrial respiratory function and protecting against oxidative damage. In vitro studies have demonstrated that taurine can normalize cellular respiration rates and improve the synthesis of key mitochondrial proteins, which are impaired in MELAS [70,71,72]. Furthermore, taurine supplementation aids in preventing the progression of neurological deficits by stabilizing mitochondrial membranes and reducing the production of ROS. Its neuroprotective effects extend to reducing excitotoxicity in neural tissues, further reinforcing its therapeutic potential in MELAS [2,29,71].

In addition to its impact on mitochondrial function, taurine has systemic benefits. These include reducing lactic acidemia, which is a hallmark of MELAS, and protecting against muscle degeneration, both of which significantly contribute to the morbidity of the syndrome. By stabilizing cell membranes and preventing calcium overload, taurine helps maintain muscle function and reduce fatigue, which are common issues in patients with MELAS [29,72]. Dosing in taurine therapy is tailored based on patient age and body weight. For pediatric patients, the recommended dosage is 200–400 mg/kg per day, divided into two or three doses, depending on body weight. For adult patients, the typical dosage ranges from 9 to 12 g per day, administered in divided doses. Taurine therapy is generally well-tolerated, although mild gastrointestinal adverse effects such as nausea and diarrhea can occur. These adverse effects are usually transient and can be managed by gradually increasing the dose or dividing it into smaller, more frequent administrations [57]. No severe adverse effects have been consistently reported, which is important and makes taurine a safe long-term option for managing MELAS symptoms [57,69,70]. Taurine’s ability to combat oxidative stress and support mitochondrial health, combined with its safety profile, positions it as a key therapeutic component in the multidisciplinary management of MELAS.

### 5.3. Diet Therapies

Dietary therapies, particularly the KD, have emerged as a potential treatment for improving mitochondrial dysfunction, especially in conditions such as MELAS. The KD works by inducing a state of ketosis, which shifts the body’s primary energy source from glucose to ketones, thus promoting more efficient ATP production in mitochondria through fatty acid oxidation. This shift is crucial in patients in whom mitochondrial oxidative phosphorylation is impaired, such as those with MELAS. Several studies have demonstrated that the KD enhances mitochondrial energy metabolism, stabilizes ATP production, and reduces oxidative stress, contributing to improved neurological function in affected patients [73,74]. Glutamate-induced deregulation of the Krebs cycle in MELAS syndrome leads to mitochondrial dysfunction, and ketone body exposure can alleviate this dysfunction by reducing glutamate levels and restoring mitochondrial function; these factors suggest potential therapeutic benefits for patients with MELAS [75]. Moreover, ketosis helps maintain synaptic function, providing neuroprotective benefits in addition to energy metabolism support [56].

The metabolic shift induced by the KD enhances mitochondrial function and helps reduce the burden of lactic acidosis, which is a hallmark of MELAS. By reducing the reliance on glycolysis, the KD helps lower lactate levels in the blood, which is crucial for preventing the progression of SLEs in these patients [76,77]. Some reports have also highlighted the potential of the KD in reducing oxidative stress markers, further supporting its role in the management of mitochondrial disorders [78]. In addition to the standard KD, modified dietary approaches such as the modified Atkins diet (MAD) and low-glycemic index treatment (LGIT) have been explored. With a fat-to-non-fat ratio of 1:1 or 2:1, MAD allows for more liberal protein and calorie intake and includes a daily carbohydrate restriction. LGIT focuses on low-glycemic index foods (≤50) without aiming for ketosis. Despite this, LGIT often induces mild ketogenesis, resulting in seizure control with minimal side effects and improved nutritional balance [77,78,79]. These diets provide similar metabolic benefits by promoting ketone body production while being more sustainable and easier to adhere to, especially for long-term treatment. MAD and LGIT have been shown to reduce seizure frequency and improve overall metabolic function in patients with mitochondrial disorders, including MELAS, with fewer adverse effects than those with the traditional KD [56,76]. Studies have confirmed that these diets, while less restrictive, can help stabilize mitochondrial energy metabolism and reduce lactic acidosis, a common feature of MELAS [77,78]. Furthermore, their more lenient structure allows for greater patient adherence, thereby making them a preferable option for patients who struggle with the strictness associated with the KD [79].

The use of modified dietary therapies such as MAD and LGIT holds particular promise for patients with MELAS because these approaches mitigate the common gastrointestinal and metabolic adverse effects of a strict KD such as constipation, elevated cholesterol, and nutrient deficiencies. The flexibility and lower risk profile of these diets make them more sustainable options for long-term management, ensuring better patient compliance and overall quality of life. Additionally, their ability to maintain stable ketone levels with fewer dietary restrictions makes them an attractive alternative for the continued management of mitochondrial dysfunction in MELAS [56,79,80].

## 6. Future Directions

Epigenetic and gene-editing therapies represent innovative approaches in the exploration of mitochondrial disease treatments, including MELAS. While epigenetic editing targets gene expression through mechanisms such as histone modifications and mtDNA methylation, its application to mtDNA remains theoretical at present. Techniques like mitochondrial-targeted TALENs (Transcription Activator-Like Effector Nucleases) have shown potential for selectively editing mtDNA in preclinical studies, offering a pathway to mitigate pathogenic mtDNA mutations such as m.3243A>G. However, significant technical challenges remain in delivering these tools to the mitochondria effectively. Although CRISPR/Cas9 has revolutionized nuclear DNA editing, its use for mtDNA editing is currently limited because of its inability to access the mitochondrial matrix. Future advancements in mitochondrial delivery systems and precision gene-editing technologies will be crucial for realizing these approaches. Nonetheless, these strategies highlight a promising direction for addressing the underlying genetic causes of MELAS, which could complement existing symptomatic therapies [7,14,31,81].

Mitochondrial replacement therapy (MRT) is another promising approach, particularly in the context of maternal inheritance. MRT involves transferring healthy mitochondria into the oocytes of mothers carrying pathogenic mtDNA mutations, thereby preventing the transmission of the disease. Although this technique has been successfully used in animal models and has entered human trials for certain mitochondrial diseases, ethical and technical challenges need to be overcome before it can be widely adopted [82]. Moreover, mitochondrial transplantation, where functional mitochondria are directly transferred into damaged tissues or organs, is an experimental approach that has shown early success in preclinical studies and holds potential for treating acute mitochondrial dysfunction in diseases such as MELAS [81].

Cannabinoids are also emerging as a potential therapeutic strategy because of their ability to improve mitochondrial function. Recent studies have demonstrated that cannabidiol enhances mitochondrial health by inhibiting mitophagy through the PINK1/PARKIN pathway, which may improve energy production and cellular function in muscle tissues. This ability to modulate autophagy and mitophagy could provide neuroprotective benefits, making cannabinoids a novel treatment option for mitochondrial diseases such as MELAS [83,84].

Bezafibrate, a peroxisome proliferator-activated receptor activator, is being explored for its ability to enhance mitochondrial fatty acid oxidation and improve energy metabolism in individuals with mitochondrial disorders, including MELAS. KH176, a vitamin E derivative, has shown promise in early clinical trials by scavenging ROS and improving symptoms such as attention, depression, and anxiety in mildly affected patients with MELAS [2,85]. Sirt1, a nuclear deacetylase, is activated by elevated cellular nicotinamide adenine dinucleotide (NAD+) levels, which deacetylate acetyl-lysine residues in proteins. Enhancing NAD+ through precursors like nicotinamide riboside or inhibiting NAD-consuming enzymes such as polyribosylpolymerase 1 has demonstrated benefits in animal models of mitochondrial diseases, highlighting the potential for further research to optimize these therapeutic strategies [85]. Also, hydrogen sulfide (H2S)-based therapies, by enhancing mitochondrial respiration and promoting cytoprotective mechanisms, show potential for improving energy production and mitigating disease progression in primary mitochondrial diseases [86].

## 7. Conclusions

The diagnosis and management of MELAS syndrome require a comprehensive, multidisciplinary approach that includes clinical evaluation, genetic testing, biochemical assays, and neuroimaging. Current management strategies focus on symptom control through supplements such as L-arginine, taurine, and dietary therapies, while addressing the multisystemic nature of the disease with a tailored, organ-specific approach. As scientists’ understanding of mitochondrial dysfunction continues to deepen, new treatments are emerging, such as mitochondrial transplantation and gene therapy, that offer the potential to address the underlying causes of MELAS [81]. The increasing focus on precision medicine, which tailors treatments to individual genetic profiles, holds promise for optimizing therapeutic outcomes and minimizing adverse effects [12]. With these advancements, more effective, targeted treatments could hopefully improve the quality of life for patients with MELAS and potentially offer curative options in the future.

## Figures and Tables

**Figure 1 biomolecules-14-01524-f001:**
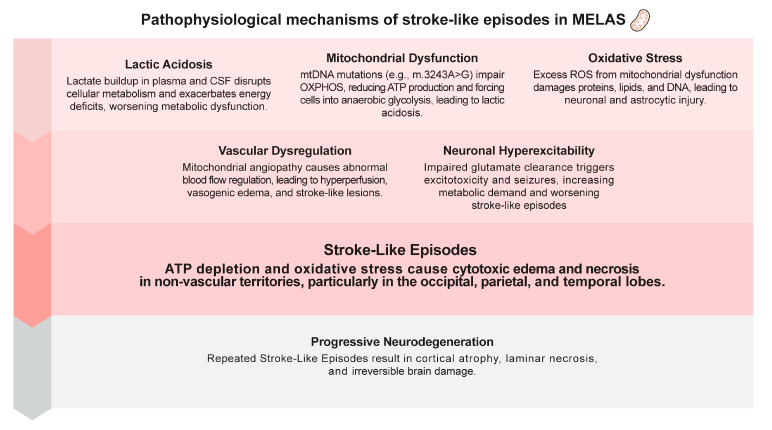
Pathophysiological mechanisms of stroke-like episodes in MELAS. ATP, adenosine triphosphate; CSF, cerebrospinal fluid; DNA, deoxyribonucleic acid; MELAS, mitochondrial encephalopathy lactic acidosis and stroke-like episodes; mtDNA, mitochondrial deoxyribonucleic acid; OXPHOS, oxidative phosphorylation; ROS, reactive oxygen species.

**Figure 2 biomolecules-14-01524-f002:**
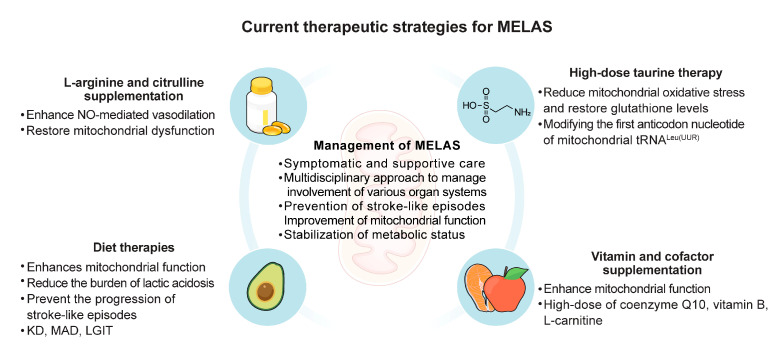
Current therapeutic strategies for MELAS. KD, ketogenic diet; LGIT, low-glycemic index treatment; MAD, modified Atkins diet; MELAS, mitochondrial encephalopathy lactic acidosis and stroke-like episodes; NO, nitric oxide.

**Table 1 biomolecules-14-01524-t001:** Key diagnostic modalities and their clinical significance in MELAS.

Diagnostic Modality	Finding	Clinical Relevance
Genetic Testing	m.3243A>G mutation in *MT-TL1* gene; other mtDNA mutations (e.g., *MT-ND5*, *MT-TH*) [1,2,8]	Confirms a MELAS diagnosis; identifies mutation load and heteroplasmy levels
Clinical Presentation	Stroke-like episodes, seizures, diabetes mellitus, hearing loss, weakness, headache, cortical vision loss [2]	Critical for the early suspicion of MELAS, especially in patients younger than 40 years with stroke-like episodes
Brain MRI	Stroke-like lesions not confined to vascular territories; cortical/subcortical hyperintensities [5,6]	Helps differentiate MELAS from ischemic strokes; detects progressive structural brain changes
Plasma or CSF Test	Lactic acidosis; elevated lactate and pyruvate levels [1,8]	Indicates systemic metabolic dysfunction; important for assessing metabolic crises
Muscle Biopsy	Ragged red fibers (RRFs); abnormal mitochondrial morphology; multiple partial defects in respiratory chain complexes, particularly complex I and/or complex IV [1,8,20]	Confirms mitochondrial dysfunction, particularly when genetic testing is inconclusive; not necessary for diagnosing MELAS because molecular genetic testing is often preferred and commonly utilized to confirm a diagnosis
MRS	Elevated lactate levels in affected brain regions indicative of impaired oxidative phosphorylation [5,6,8]	Supports a diagnosis of mitochondrial dysfunction and energy failure
Urine Test	Elevated urinary lactate; high mutation load in urinary epithelial cells [1]	Noninvasive method to assess mutation load in mtDNA; it is correlated with disease severity; not necessary for diagnosing MELAS
EEG	Generalized slowing, epileptiform discharges during stroke-like episodes [13,35,36]	Detects seizures and abnormal brain activity; useful for monitoring epileptic activity; unnecessary for diagnosing MELAS
Cardiac Evaluation	Hypertrophic cardiomyopathy, arrhythmias on ECG or echocardiogram [40,43]	Assesses cardiac involvement; it is a significant source of morbidity in patients with MELAS; unnecessary for diagnosing MELAS
Endocrine Evaluation	Insulin-dependent diabetes mellitus [45,46]	Important for identifying early endocrine involvement, which may precede neurological symptoms; not necessary for diagnosing MELAS

Abbreviations: ECG, electrocardiogram; EEG, electroencephalogram; MELAS, mitochondrial encephalopathy lactic acidosis and stroke-like episodes; MRS, magnetic resonance spectroscopy.

## Data Availability

The data that support the findings of this study are available from the corresponding author upon reasonable request.

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
