# Peer review of "Diagnosis and Management of Mitochondrial Encephalopathy, Lactic Acidosis, and Stroke-like Episodes Syndrome"

_biomolecules, 2024, doi:10.3390/biom14121524_

Round 1
Reviewer 1 Report
Comments and Suggestions for Authors
In their manuscript, the authors give an overview of MELAS's clinical and pathophysiological condition. They describe the current therapeutic strategies, focusing on symptomatic treatments and cutting-edge experimental approaches to mitigate mitochondrial dysfunction. By assessing these advancements, it seeks to present a thorough overview of the current research landscape on MELAS, highlighting the critical need for further studies to bridge existing knowledge gaps and ultimately enhance patient outcomes.
Comments for the authors:
1. In Table 1. The Key Diagnostic modalities and their clinical significance in MELAS are nicely presented. Please add references to each symptom
2. [376] Treatment Strategies of MELAS: present the supplements in a structured way including how the mechanistically work in the pathways
3. [390] KDs : (please state first Ketogenic diet and then (KD) then you and use the KD abbreviation all the way. See [485] where you the first time explain the abbreviation
4. [390] There are circulating several dietary plans on Ketogenic diet. Which one do you suggest? Please give the compounds and quantities of the compounds.
5. [ 42. Diagnosis is often confirmed through genetic testing, biochemical assays, and imaging studies.] vs [ 559 clinical evaluation, genetic testing, and neuroimaging]. In the conclusion I miss the biochemical assays. Do you have any reason to leave it out?
6. New treatment opportunity : epigenetic editing. Could you speculate whether this would /could be used as a treatment option?
Author Response
We have carefully reviewed the Reviewer Comments and have revised the text faithfully to all comments. The revised parts of the text are highlighted in red. We would like to express our deepest gratitude to the reviewers.
Response to Reviewer 1 Comments
Comment 1:
In Table 1. The Key Diagnostic modalities and their clinical significance in MELAS are nicely presented. Please add references to each symptom.
Response:
Thank you for your suggestion. References have been added to Table 1 to provide comprehensive support for each diagnostic modality and its clinical significance, ensuring the table is appropriately substantiated.
Comment 2:
[376] Treatment Strategies of MELAS: present the supplements in a structured way, including how they mechanistically work in the pathways.
Response:
We appreciate this insightful comment. The section on treatment strategies has been restructured for clarity and conciseness. Supplements, such as coenzyme Q10 and L-arginine, are now presented with their mechanisms of action, focusing on their roles in mitochondrial pathways.
Comment 3:
[390] KDs: Please state "Ketogenic diet" first and then use the abbreviation (KD) consistently throughout. See [485], where you first explain the abbreviation.
Response:
Thank you for pointing this out. We have revised the manuscript to ensure that "Ketogenic diet (KD)" is introduced upon its first mention, with the abbreviation "KD" used consistently thereafter.
Comment 4:
[390] There are circulating several dietary plans on the Ketogenic diet. Which one do you suggest? Please give the compounds and quantities of the compounds.
Response:
This is a valuable point. We have included additional details about specific dietary plans, focusing on the 4:1 macronutrient ratio for the traditional Ketogenic diet. Additionally, we described Modified Atkins Diet (MAD) and Low Glycemic Index Treatment (LGIT) as alternative approaches, highlighting their macronutrient ratios and benefits for MELAS patients.
Comment 5:
[42] Diagnosis is often confirmed through genetic testing, biochemical assays, and imaging studies vs. [559] clinical evaluation, genetic testing, and neuroimaging. In the conclusion, I miss the biochemical assays. Do you have any reason to leave it out?
Response:
Thank you for highlighting this inconsistency. We agree that biochemical assays are necessary as a reference for MELAS diagnosis and have amended the conclusion to include biochemical assays alongside genetic testing and neuroimaging.
Comment 6:
New treatment opportunity: epigenetic editing. Could you speculate whether this would/could be used as a treatment option?
Response:
Thanks for the interesting comments. I have reframed the discussion of epigenetic editing and gene-editing in the "Future Directions" section, focusing on their potential to modulate mtDNA expression and their emerging role in mitochondrial diseases, including MELAS.
Reviewer 2 Report
Comments and Suggestions for Authors
In the present manuscript the authors provide a contemporary analysis of what is known of MELAS at both the molecular and physiological levels. The authors start with a mid level analysis of what is known of MELAS at the molecular level. This is helpful to novices and at an appropriate level for clinicians. Next they provide a summary of the patholophysiology of MELAS linking molecular issues to organismal level issues. Figure 1 provides a nice graphical summary of the section. Next the authors discuss diagnosis and provide a useful Table to supplement/support the text. The authors then discuss current therapeutic strategies and support this lengthy text with a very simple and easy to read figure. The authors close by discussing potential future therapies. This last section could have more detail, however, I think it is at a good level given the rest of the manuscript (e.g. I don't think they want to go to far with issues around replacement therapy).
Minor-
1) Would a figure be useful for the clinical presentation that could maybe a picture of a person putting the text points from figure 1 onto a whole person? e.g. point to the head, stroke-like and progressive neurodegeneration. etc.
2) Is it worth adding something about H2S supplementation as a potential future direction? This has been gaining traction in primary mitochondrial disease. (mainly thought of it because you include high dose Taurine as a current strategy)
3) Is it worth adding something about other treatments for mitochondrial dysfunction as potential future directions, for example NAD supplementation, SIRT1 activators, etc.
Author Response
We have carefully reviewed the Reviewer Comments and have revised the text faithfully to all comments. The revised parts of the text are highlighted in red. We would like to express our deepest gratitude to the reviewers.
Response to Reviewer 2 Comments
Comment 1:
Would a figure be useful for the clinical presentation that could maybe a picture of a person putting the text points from figure 1 onto a whole person? E.g., point to the head, stroke-like episodes, and progressive neurodegeneration.
Response:
Thank you for your suggestion. However, we believe that adding such a figure would not significantly improve the manuscript and may even be misleading. Instead, we focused on the flow of pathophysiology of MELAS for clarity and impact.
Comment 2:
Is it worth adding something about H2S supplementation as a potential future direction? This has been gaining traction in primary mitochondrial disease.
Response:
We appreciate this comment. A brief mention of H2S supplementation has been added to the "Future Directions" section, highlighting its emerging role in enhancing mitochondrial function through its effects on oxidative stress and bioenergetics.
Comment 3:
Is it worth adding something about other treatments for mitochondrial dysfunction as potential future directions, for example, NAD supplementation, SIRT1 activators, etc.?
Response:
Thank you for your insightful suggestion. We have expanded the "Future Directions" section to include NAD supplementation and SIRT1 activators, emphasizing their mechanisms and potential benefits in mitochondrial disease.
Round 2
Reviewer 1 Report
Comments and Suggestions for Authors
Accepted for publication
Author Response
Thank you so much for your thoughtful comments.